# Variable-Constrained Model Predictive Control of Coordinated Active Power Distribution for Wind-Turbine Cluster

**Zhenyu Chen *** **, Jizhen Liu, Zhongwei Lin and Chenzhi Qu**

State Key Laboratory of Alternate Electrical Power System with Renewable Energy Sources,
School of Control and Computer Engineering, North China Electric Power University, Beijing 102206, China;
ljz@ncepu.edu.cn (J.L.); lzw@ncepu.edu.cn (Z.L.); qchenzhi@outlook.com (C.Q.)
**\*** Correspondence: czy_ac@outlook.com

**Abstract:** In this paper, a wind-turbine active power control strategy is proposed from the cluster level to optimize the active power set-point for each wind turbine in a specific cluster. The wind turbine power tracking characteristic is described as an inertial link to establish the power tracking predictive model, and the model predictive control (MPC) method is used to optimize the cluster power demand and output. Time-varying constraints are proposed to coordinate the power output for different time-scale wind turbines and sustain that the cluster has enough fast-tracking capacity when the cluster power demand changes. Under different scenarios, the proposed strategy is tested to verify the effectiveness in improving the power output stability and frequency support ability.

**Keywords:** wind turbine; active power control; wind-turbine cluster; model predictive control; frequency regulation

## 1. Introduction

With the development of modern energy industry, renewable energy gradually increases its proportional in the energy field, of which the wind energy is also developed rapidly due to its clean and environmentally friendly features. Usually, the wind energy is captured and converted by the wind turbine generator systems (WTGSs). Over the past few decades, much research and developments have been done in wind turbine control to achieve high quality generator power output and decrease cost and mechanical loads [1,2]. The wind turbine is mainly controlled differently in regions. During the cut-in wind speed and rated speed region, the maximum power point tracking is the most important control object [3], and the output power will be limited to the rated condition when the wind speed is above the rated speed but below the cut-out speed [4].

In industrial fields, WTGSs within an area are always controlled under a wind farm. The wind farm calculates the available power for the whole farm and uploads to the power grid dispatching center, which also takes the wind farm power set-point from the dispatching center in the mean time and schedules the active power set-points for all inner WTGSS. All WTGSs output power are connected together as the wind farm power output to follow the wind farm power demand. To balance the power supply and demand of the electric grid, the power set-point for the wind farm is usually given in two modes, i.e., free generation mode and power limitation mode. At the free generation mode, each unit runs freely to track the optimal operating point under current wind speed. The power set-point for the wind farm is usually set below the optimal operating condition at power limitation mode, and the wind farm reserves spare capacity from the current maximum output [5]. The reserved capacity can be used in electric power system peak load and frequency regulation which is called *Deloading Control* [6].

Many active power control strategies have been studied in the wind farm to distribute the power set-point for each unit under the power limitation mode. Ref. [7] proposed a two-time-level wind farm active power control to improve the wind power output, where a distributed model predictive control strategy is introduced and ADMM (Alternating Direction Method of Multipliers) algorithm is used to speed up the solution process. Refs. [8,9] proposed a wind farm distributed model predictive control with clustering-based piece-wise affine wind turbine model on the basis, complete wind turbine model is used and the wind farm is set to track the active power output as the conventional strategy while reducing the wind turbine loads. Another farm control load-reduction strategy is introduced in [10], of which the power tracking error and loads' conditions are combined in the cost function and solved by a centralized model predictive controller. When the wind farm is connected into the power grid, voltage will be an important parameter that needs to be considered. Ref. [11] proposed a VSC-HVDC (Voltage Source Converter based High Voltage Direct Current Transmission) connected wind farm active power control when the voltage variation is taken into consideration to improve the wind farm voltage performance. Since WTGSs in a wind farm are growing in number, the optimization problem from the wind farm level will also increase in orders and the solving time required. In such a case, the decentralized strategies are also used in wind farm power and voltage control process. The related research can be referred to from [12,13].

As an important part in electric power grid, the wind farm is also controlled to support the peak and frequency for the electric power grid. According to the time scale, the frequency regulation of power system is mainly divided into different levels [14]. In the primary frequency regulation (PFR), the power generation unit takes the power system frequency error as the input, and the unit power output is adjusted by the unit control system to maintain the power system frequency stabilized at the target. The secondary frequency regulation (SFR) is also marked as AGC (Automatic Generation Control), which controls the output power for different power generation units to meet the frequent changing power demand and keeps the power system under economic operations. The reviewed strategies in the above paragraph were focused on the wind farm AGC, while several other control methods like the *Droop Control* and *Inertial Control* are introduced in WTGS to involve wind energy in the PFR [15]. The kinetic energy stored in the WTGS rotor will be released under frequency drop conditions to support the grid frequency. A new wind farm PFR strategy is introduced in [16], firstly, the PFR strategy is first designed for one WTGS, and then the proposed strategy is implemented into the wind farm by the participation-factor-based wind turbine control strategy. In this way, the wind farm under the proposed strategy could have the PFR ability while avoiding individual turbine control.

Although the previously strategies are useful and efficient, there still exist several potential parts which can be further improved in detail. Firstly, a wind farm level exists to arrange all the WTGSs according to the geographical location, and the power control and frequency regulation can be controlled and optimized on the wind-farm level. The increase in number of WTGSs in the wind farm may cause calculation problems during optimization. For higher reliability, a wind-turbine cluster level can be selected as a middle layer between the single wind turbine and wind farm level to control and optimize the wind turbine active power output. Secondly, in the industrial field, it is difficult for the wind turbine to output all the available power due to the power set-point for the cluster or wind farm is usually lower than its available power. Though the reserved capacity for a wind turbine may be not sufficient enough, the reserved capacity for a wind-turbine cluster is still impressive and can be used to stabilize the cluster power output and support grid frequency. In China, some active power limiting strategy has begun to implement in wind farms to keep wind farms have reserved power for power regulation and frequency support, such as Gansu province. There will be large freedom degrees for the wind power for stabilized power output and supporting the power system stability. Thirdly, a complete wind turbine model may be helpful when considering the turbine load characteristic, but a simplified model could be useful to reduce the problem order in the optimal wind farm active power control. The dynamic characteristic for a wind turbine will be changed under continuous operations and cause a difference between wind turbines. Such property should also be taken into consideration

during active power control. In addition, many model predictive control (MPC) related strategies are used in wind farm active power distribution. From the control theory perspective, there still exist potential improvements to further develop the control method. Like the state assessment based variable weight MPC strategy proposed in [17], the further improvement based on current model predictive control framework is also a focused problem. It should be noted that the wake effect is not considered in this paper, but the proposed strategy still can be used in a waked wind farm. Reasons for this will be explained in the main text.

In this paper, a wind turbine active power control strategy is proposed to optimize the active power set-point from a wind-turbine cluster level controller. The power tracking dynamic characteristic for each wind turbine is modeled as a first-order inertial link to describe the wind-turbine cluster power tracking dynamic response, and the model predictive control strategy is adopted to optimize the power tracking error between the cluster power demand and output. The rest of this paper is organized as follows: in Section 2, the simplified NREL 5MW wind turbine model is introduced. Section 3 discusses the wind turbine power tracking dynamic characteristics, which are combined into the active power model predictive controller in Section 4. Time-varying constraints are proposed in Section 5 to limit and balance the power set-point for each wind turbine. In Section 6, the proposed strategy is tested under different cases. Section 7 concludes this paper.

## 2. Simplified NREL 5MW Wind Turbine

The WTGS is essentially a nonlinear model with multiple variables and controllers to capture the wind energy and transform it into the electric power. The wind-turbine cluster is defined as a cluster of WTGSs which can be controlled and scheduled uniformly. The wind-turbine cluster can represent a wind farm with multiple equivalent wind turbines, and can also be used as an intermediate level including several WTGSs in one wind farm to stratify the wind farm and every WTGSs. Essentially, wind-turbine cluster take WTGSs as minimum components. Research in this paper is implemented based on simplified *NREL* 5MW wind turbine provided within the *SimWindFarm* simulate toolbox. The *SimWindFarm* toolbox is developed as part of the Aeolus FP7 project and is aimed at providing a fast wind farm simulation environment for development of wind farm control algorithms. In this section, basic concept about NREL 5MW WTGSs and the used controller in *SimWindFarm* are introduced before further wind-turbine cluster's controller design.

### 2.1. WTGS Model

A variable-speed wind turbine generally consists of an aeroturbine, a gearbox and a generator. The aerodynamic power captured by the rotor is given by the nonlinear expression as:

$$P_r = \frac{1}{2}\rho\pi R^2 C_p(\lambda, \beta)V^3,\tag{1}$$

where $\omega_r$ is the rotor speed, $R$ is the rotor radius, and $\rho$ is the air density. The power coefficient $C_p$ is a nonlinear function defined based on the blade pitch angle $\beta$, and the tip-speed ratio $\lambda$, where the tip speed ratio is:

$$\lambda = \frac{\omega_r R}{V}.\tag{2}$$

$C_p$ is used to establish the aerodynamic model for the capturing ability of wind energy. For different wind turbines, the polynomial expressions of $C_p$ are usually with a similar form but different parameters. The adopted $C_p$ for the simulated wind turbine can be referred to [18]. Similar to the aerodynamic power, the aerodynamic torque is defined as follows:

$$T_r = \frac{1}{2}\rho\pi R^3 C_q(\lambda, \beta)V^2,\tag{3}$$

where

$$C_q(\lambda, \beta) = \frac{C_p(\lambda, \beta)}{\lambda}. \tag{4}$$

The dynamic response of the wind turbine rotor driven at a speed $\omega_r$ by the aerodynamic torque $T_r$ is shown to be

$$J_r \dot{\omega}_r = T_r - T_{ls} - K_r \omega_r. \tag{5}$$

The low-speed shaft torque $T_{ls}$ acts as braking torque on the rotor. It results from the torsion and friction effects due to the difference between $\omega_r$ and low speed shaft speed $\omega_{ls}$:

$$T_{ls} = K_{ls}(\theta_r - \theta_{ls}) + B_{ls}(\omega_r - \omega_{ls}), \tag{6}$$

where $\theta_r$ and $\theta_{ls}$ are the angular positions of the rotor and shaft, respectively. The generator is driven by the high-speed shaft torque $T_{hs}$ and braked by the generator electromagnetic torque $T_{em}$:

$$J_g \dot{\omega}_g = T_{hs} - T_{em} - K_g \omega_g. \tag{7}$$

Assuming an ideal gearbox with transmission ratio $n_g$, the following equation can be established

$$n_g = \frac{T_{ls}}{T_{hs}} = \frac{\omega_g}{\omega_{ls}} = \frac{\theta_g}{\theta_{ls}}. \tag{8}$$

It should be noticed that the wind turbine introduced is modeled with a flexible driveshaft, but the simulated wind turbine in FAST can be developed with more DOFs (degree of freedoms) like the flapwise and edgewise blade mode and fore-aft and side-to-side tower bending-mode, but the simplified WTGS in *SimWindFarm* is designed with a few DOFs like generator control and flexible drivetrain. Such simplification could simplify the simulation process and reduce the running time while preserving the basic characteristics of wind turbines.

In the baseline NREL 5MW wind turbine model, the generator model is usually user defined, but a simple first order generator model is used in the *SimWindFarm* benchmark. The simple generator model can be expressed as:

$$\dot{T}_{em} = \frac{1}{t_{gen}} \left( \frac{P_{set}}{\omega_g} - T_{em} \right) \tag{9}$$

in which $P_{set}$ is the active power set-point for the WTGS and $t_{gen}$ is the generator time constant.

*2.2. Wind Turbine Control Strategy*

As can be seen in Equation (9), the generator takes an active power set-point as an input. Generator speed is used to regulate the WTGS operations. A baseline control strategy for NREL 5MW wind turbine is provided in [19]. The controller in *SimWindFarm* is designed based on the baseline control strategy with little modifications. When the wind speed is lower than the rated wind speed, a power/generator-speed relationship lookup table is used, the WTGS is controlled by the generator speed to track the maximum power point, and the blade pitch is kept at zero. When the wind speed is above the rated speed, the generator speed is set to the rated speed, and the blade pitch angle is controlled under a gain-scheduled PI controller to maintain the wind turbine power output at rated operating point. Detailed WTGS controllers in *SimWindFarm* toolbox can be referred to [20]. In addition, a region check function is used in WTGS control to make sure the WTGS could work under partial load conditions, the blade pitch control will also work under the partial load condition that the WTGS power will also have stable output stable the power set-point, even though the power set-point is lower than the current operation point.

*2.3. Available Power Calculation for WTGS*

As shown in Equation (1), the aerodynamic power captured by the rotor is a nonlinear function of blade pitch angle and tip speed ratio. Under the baseline control strategy used in *SimWindFarm*, the WTGS is controlled to achieve the appropriate blade pitch angle and tip speed ratio under different wind conditions. In other words, the WTGS with controllers can be assumed to be able to track appropriate operating points defined by the wind speed, thus the WTGS available power can be determined once the wind speed is achieved. The available power calculation is simplified as a static nonlinear function. During simulations, the nonlinear function is implemented by a lookup table, and the implemented lookup table is shown in Figure 1. Detailed parameters can be obtained from [21].

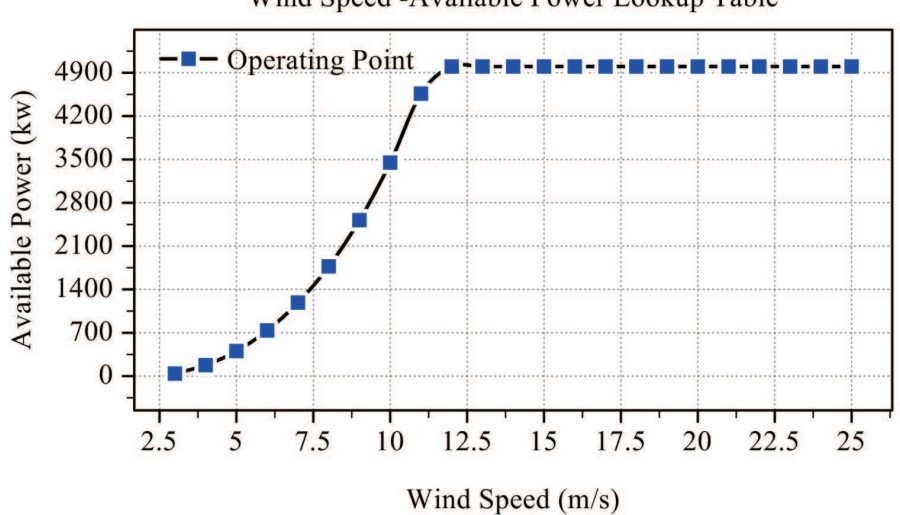

**Figure 1.** Wind speed—available power lookup table.

Considering that the WTGS control strategy covering the whole control region includes both full load and partial load conditions, the WTGS power output can be adjusted from zero to the available power flexibly. From the wake effect perspective, in this paper, the wind speed for each unit is assumed to be easily measured, this basic assumption already includes the wake effect that the measured wind speed is the actual wind speed acting on the wind turbine. Once the rear unit is affected by the wake of the front unit, the measured wind speed for rear unit is the waked wind speed.

## 3. Active Power Tracking Dynamic Characteristics of Wind-Turbine Cluster

From the control design perspective, the blade pitch controller, the generator speed/torque controller and the yaw controller are working in coordination to make sure the WTGS functional. However, from the energy transfer perspective, a WTGS usually receives active power orders from the active power controller in the wind-farm level, and then adjusts the inside controllers to make sure that the generator power output catches up with the active power demand. In such a situation, the inner controllers can be regarded as WTGS inherent modules. The dynamic characteristic of power tracking for a WTGS can be reduced to a one-order inertial link from the active power demand to the generator power output, for which the input is the WTGS active power set-point from the wind farm and the output is the generator power output. The WTGS power tracking dynamic characteristic can be expressed as follows:

$$P_i^{out} = \frac{1}{T_i s + 1} P_i^{set},\tag{10}$$

where *i* represents the *i*-th wind turbine. The inertia time constant $T_i$ can be easily obtained through the black box system identification and least square fit as shown in Figure 2.

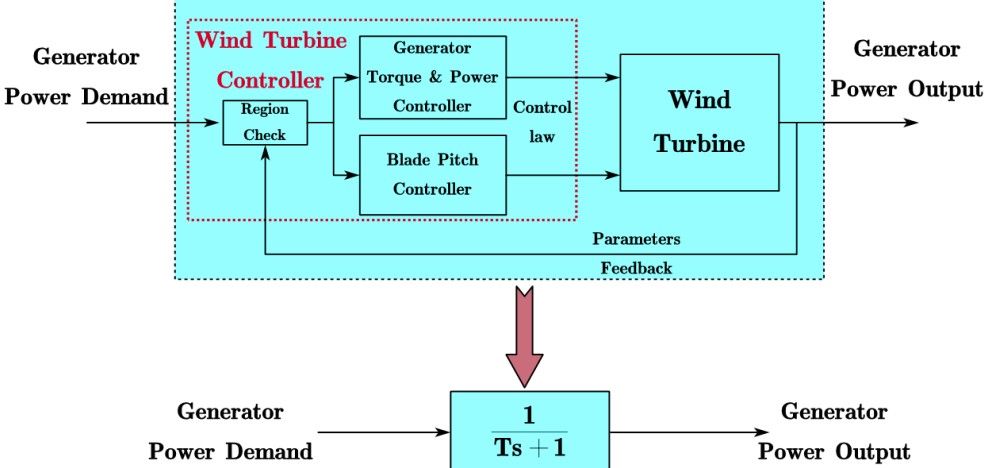

**Figure 2.** Wind turbine power tracking dynamic characteristics.

Furthermore, the WTGS active power tracking dynamic characteristic in Equation (10) can be rewritten as a state-space model:

$$\begin{cases} \dot{x}_i = -\frac{1}{T_i}x_i + \frac{1}{T_i}u_i, \\ y_i = x_i, \end{cases} \tag{11}$$

in which the input $u_i$ is the $i$-th wind turbine active power set-point $P_i^{set}$, the state $x_i$ and the output $y_i$ are the $i$-th wind turbine generator active power output $P_i^{out}$ tracking the active power set-point, respectively.

Under such a circumstance, during the active power distributing process in a wind-turbine cluster, the cluster active power output is the combination of each inner wind-turbine active power, which is affected by the active power tracking dynamic characteristic of each WTGS in the cluster. Assuming that there are a total of $n$ WTGSs in the cluster, the cluster active power output can be expressed as follows:

$$\begin{cases} \dot{x} = Ax + Bu, \\ y = Cx, \end{cases} \tag{12}$$

in which the input variable is defined as the active power set-point for each WTGS with $u = \left[P_1^{set}, P_2^{set}, \ldots, P_n^{set}\right]'$. The state is the active power output for each WTGS with $x = \left[P_1^{out}, P_2^{out}, \ldots, P_n^{out}\right]'$, and the output $y$ is the output active power $P_{cluster}^{out}$ for the specific wind-turbine cluster. The state space model parameters are given as follows:

$$A = \begin{bmatrix} -\frac{1}{T_1} & & & \\ & -\frac{1}{T_2} & & \\ & & \ddots & \\ & & & -\frac{1}{T_n} \end{bmatrix}_{n \times n},$$

$$B = \begin{bmatrix} \frac{1}{T_1} & & & \\ & \frac{1}{T_2} & & \\ & & \ddots & \\ & & & \frac{1}{T_n} \end{bmatrix}_{n \times n}, \quad C = [1, 1, \ldots, 1]_{1 \times n}.$$

Clearly, Equation (12) provides a feasible way to model the wind-turbine cluster active power tracking as a dynamic system, and the modern control method can be adopted to control the wind-turbine cluster active power dynamic process and optimize the cluster turbines operations. The wind-turbine cluster level active power control strategy will be introduced in the next section. In

order to simplify the notations, one WTGS is marked as a *unit* and a wind-turbine cluster is marked as a *cluster* briefly in the following analysis, and all the *power* used later refers to the active power if not specially marked.

## 4. Active Power Model Predictive Control of Wind-Turbine Cluster

In this section, the cluster active power MPC method is proposed based on the WTGS power tracking dynamic characteristics established in the previous section, and the cluster is controlled to track the power set-point from the wind farm or the electric grid.

### 4.1. Predictive Model

The cluster active power tracking state-space model in (12) can be further discretized through the zero-order holder method into the following discrete state space model:

$$\begin{cases} x(k+1) = A_d x(k) + B_d u(k), \\ y = C_d x(k), \end{cases} \tag{13}$$

where the matrix parameters $(A_d, B_d, C_d)$ are the discretized version from $(A, B, C)$ system. Set the initial time as $k$ and the prediction horizon as $N$; then, the predictive model can be reorganized as follows:

$$X(k) = F_x x(k) + G_x U(k), \tag{14}$$

in which

$$X(k) = \begin{bmatrix} x(k+1) \\ \vdots \\ x(k+N) \end{bmatrix}, \ U(k) = \begin{bmatrix} u(k) \\ \vdots \\ u(k+N-1) \end{bmatrix},$$

$$F_x = \begin{bmatrix} A_d \\ \vdots \\ A_d^N \end{bmatrix}, \ G_x = \begin{bmatrix} B_d & \cdots & 0 \\ \vdots & \ddots & 0 \\ A_d^{N-1} B_d & \cdots & B_d \end{bmatrix}.$$

Since the state variables $x$ are the active power output for each unit at different predict time step, the cluster power output can be calculated directly without any further prediction.

### 4.2. Cost Function

The cost function is defined as follows:

$$J = \sum_{t=1}^{N} \left| P_{cluster}^{set} - \sum_{i=1}^{n} P_i^{t,out} \right|, \tag{15}$$

in which $P_{cluster}^{set}$ is the active power set-point for the wind-turbine cluster, $P_i^{t,out}$ is the output active power of the *i*-th unit at the predictive time step $t$, $n$ is the total number of units considered in the cluster, and $N$ is the predictive time horizon. Because the time scale of the predictive horizon is much smaller than the cluster power set-point changes, the power demand or the set-point can be regarded with no changes during one predictive horizon, so the $P_{cluster}^{set}$ remains constant during one entire predictive period. Each unit power output at one predictive step are summarized together as the cluster power output, and the absolute value of difference between the cluster power set-point and output at each predictive time steps are calculated and summarized as the cost function to be minimized. The cost function is not a quadratic programming problem commonly used, but its structure is not complicated and can be easily solved by the commercial optimization solver, such as *Mosek* or *gurobi*. Simulations in this paper solve the optimization problem through the *fmincon* function provided within *Matlab* (2016a, Mathworks, Nedick, MA, USA, 2016).

### 4.3. Constraints

Several conventional constraints are still necessary during the cost function minimization, which are listed as follows:

$$\sum_{i=1}^{n} P_i^{set} = P_{cluster}^{set},\qquad(16)$$

$$0 \le P_i^{set} \le P_i^{avi}.\qquad(17)$$

For the entire cluster, Equation (16) guarantees that the cluster power of all inner units equals the cluster power demand, and Equation (17) restricts the power set for each unit within a feasible interval that each unit has the ability to track its power setting. However, the traditional studies mainly adopt the same unit dynamic characteristic, and the different current wind speeds for all separate units play the key role for the optimal power distribution. Once the dynamic characteristics of wind turbine are considered, there still exist potential improvements for the constraints, which will be explained in the following case study.

### 4.4. Initial Test

Based on the above discussion, an MPC strategy is constructed for the active power distribution of the wind-turbine cluster. As follows, the proposed strategy is tested preliminarily under a four-unit cluster to verify the feasibility. During simulations, wind turbines are differently adjusted to represent the dynamic characteristic difference. Then, the inertia time constants of simulated wind turbines are identified though the *ident* system identification toolbox in *Matlab*. The inertia time constant $T$ for four simulated wind turbines are identified as $T_1 = 0.09\,\text{s}$, $T_2 = 0.125\,\text{s}$, $T_3 = 0.20\,\text{s}$ and $T_4 = 0.3333\,\text{s}$, respectively. A conventional active power control strategy is used to compare the effectiveness, which can be expressed as:

$$P_i^{PD} = \frac{P_{cluster}^{set}}{P_{cluster}^{avi}} P_i^{avi}.\qquad(18)$$

The conventional strategy is also referred to as the proportional distribution (PD) method, which calculates the output coefficient for the wind farm and each unit to guarantee that the controlled power output meets the power demand. The compared output figures are shown as follows.

In Figure 3, the black curve is the cluster power set-point, the blue curve represents the cluster power output under the PD strategy, while the red curve is with respect to the power output under the MPC strategy. In Figure 4, the black curves represent the available power, while the blue curves represent the power output for each unit respectively, which are corresponding to constraint (17). Clearly, the cluster power output through MPC is much more stable than the conventional PD strategy, and the output power tracking performance has also been improved significantly.

Although the power output under MPC fluctuates around at about $60\,s$, but the amplitude is not dramatic and the output returns to normal within a short time period. However, the power set-point for each unit in Figure 4 shows a little deficiency in the current control strategy. The inertial time constant for each unit introduced in the previous paragraph shows that unit 1 has the best dynamic characteristic and fastest power set-point tracking ability, and unit 4 has the worst dynamic characteristic in contrast. Such characteristics can be clearly reflected in Figure 4. In particular, units 1 and 2 contribute most of their available capacity to support the cluster power output for tracking the power set-point. The power contributed by units 3 and 4 are not very much in the low-load phase, and increase their power outputs only when the cluster power set-point is large. This leads to a serious problem for the power regulation flexibility of the cluster, i.e., how to coordinate the small-inertia units and large-inertia ones for better power demand tracking with enough active power regulation flexibility left?

In fact, for a sensible cluster power control strategy, small-inertia units should play a leading part when the cluster power demand changes rapidly to achieve fast tracking of cluster power set-point.

When the quick tracking and adjustment is completed, some power output will be handed over from the small-inertia turbines so as to maintain the adequate power regulation capacity of the cluster, which should be filled by the less rapid units. In such a way, there could be enough margin left for the future cluster power set-point changes for quicker tracking and adjustment. To realize this idea, a simple but efficient strategy for MPC with time-varying constraints will be proposed in the next section.

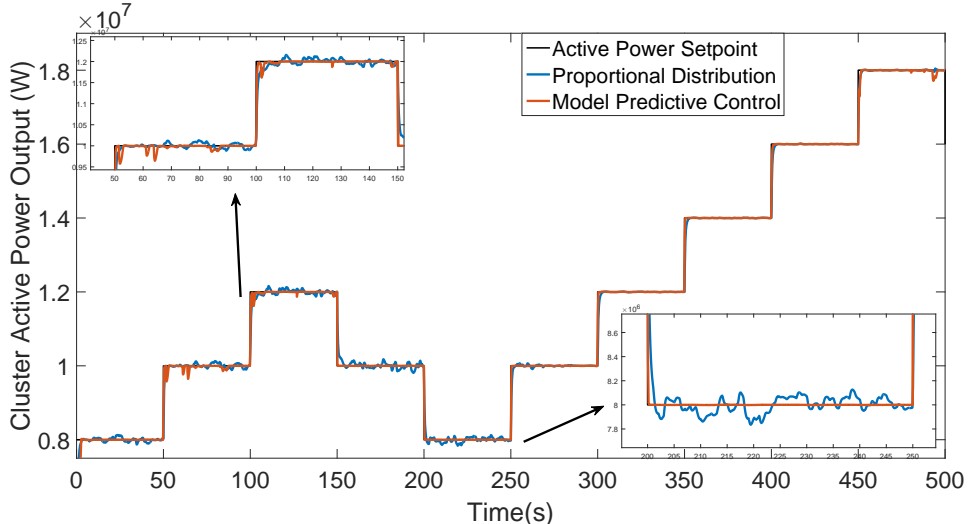

**Figure 3.** Active power output of wind-turbine cluster under step power demand.

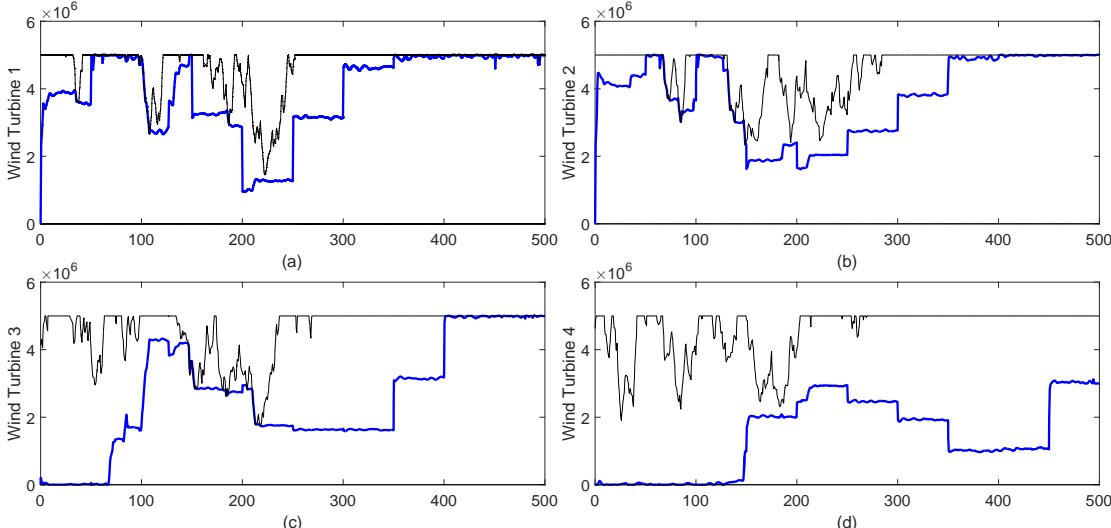

**Figure 4.** Active power output of each wind turbine under step power demand, sub-figures a, b, c, d are the active power output for wind turbine 1, 2, 3, 4 respectively.

## 5. Time Varying Constraints

To make sure the small time-scale units taking the leading part in the fast tracking of power set-points, a time-varying constraint is applied instead of (17), which can be expressed as:

$$P_i^{lb}(t) \leq P_i^{set} \leq \min \left\{ P_i^{ub}(t), P_i^{avi} \right\}, \tag{19}$$

$$P_i^{lb}(t) = \begin{cases} \left[ (1-a) - b \times \dfrac{t_0 - t}{t_0} \right] \times P_i^{PD} & , \quad t < t_0, \\ (1-a) \times P_i^{PD} & , \quad t \geq t_0, \end{cases}$$

$$P_i^{ub}(t) = \begin{cases} \left[ (1+a) + b \times \dfrac{t_0 - t}{t_0} \right] \times P_i^{PD} & , \quad t < t_0, \\ (1+a) \times P_i^{PD} & , \quad t \geq t_0, \end{cases}$$

in which $a$ and $b$ are constants satisfying $0 \leq a + b \leq 1$, $P_i^{PD}$ is the $i$-th unit power set-point under a conventional PD strategy, $t$ is the time length since the cluster power set-point changes to current time step, and $t_0$ is the constraints contraction time. Constants in (19) can be adjusted according to the practical requirements. Set $P_i^{PD}$ constant as 1 and the time varying constraints can be shown as in Figure 5.

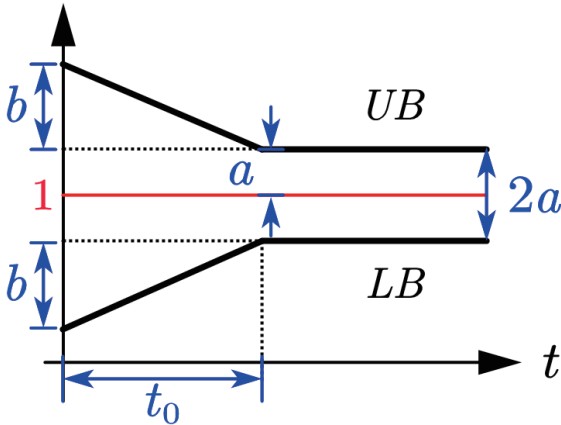

**Figure 5.** Time varying constraints.

Once the cluster set-point changes, the trigger time is reset to record the change time and stays constant in the next period. The difference between the current time and the trigger time is the calculate time $t$. Once the cluster power set-point is received with step changes, the $i$-th turbine power is set within a constrained available interval $[(1 - a - b) \times P_i^{PD}, (1 + a + b) \times P_i^{PD}]$ and gradually converges over time, after $t_0$ seconds, the available interval converges to its smallest range that the upper bound is $a$ higher than the proportional distribution method and the lower bound is $a$ lower. The gap between the upper and lower boundaries is used to overcome the power fluctuations caused by inaccurate models and inaccurate wind speed. In other words, the fast time-scale units will take the leading part when the cluster power set-point changes to meet the new set-point, and the feasible power set-point of each unit will gradually reduce so that the fast time scale units will gradually hand over the excessive power to the less rapid units until the power output for each turbine restored to a minimum bound close to the proportional distribution after $t_0$ seconds. In addition, $a + b$ is set within 0 and 1 to ensure the lower bound for one unit power output is always above zero and all units participate in the cluster power production. In addition, due to the different inertia time constant between units, the cluster power output may fluctuate during the process when restraining the units' available interval, the equality constraint in (16) is changed with a small extra range as follows:

$$(1 - c) \times P_{cluster}^{set} \leq \sum_{i=1}^{n} P_i^{set} \leq (1 + c) \times P_{cluster}^{set}. \tag{20}$$

Thus, a time-varying constrained MPC is established for the active power distribution of the wind-turbine cluster. In the following tests, constants in the time varying constraints are set as $a = 10\%, b = 40\%, c = 2\%$, and $t_0$ is 30. Considering that the wake effect is included during the available power calculation, the proposed strategy can be used for a waked wind farm although the detailed wake effect is not modeled and explained. In other words, the proposed strategy doesn't take

wake effect into consideration during controller design. Moreover, the proposed controller will be tested under several scenarios in the next section.

## 6. Case Study

In this section, the proposed strategy is tested under different scenarios to verify the effectiveness.

### 6.1. Four-Units Cluster Robustness Test

In the previous simulations, the unit available power is obtained based on the lookup table related to the measured wind speed. However, the predictive wind speed usually adopted in the real industrial field, which may introduce the external prediction error and disturbance to the cluster instructions. In order to testify the robustness of the proposed cluster-level active power control strategy, a bounded Gaussian white noise is added on the measured wind speed every few seconds, which will influence both the predictive available active power and the units' power set-point. The tested units are the same as in Section 4.4. The simulation result are shown as in Figures 6 and 7.

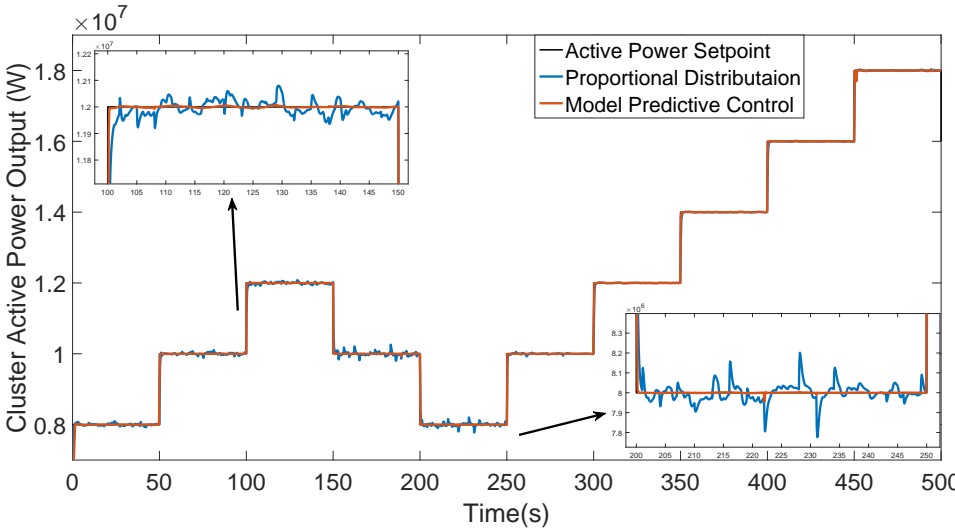

**Figure 6.** Four-units cluster robustness test—cluster power output (measured wind speed with Gaussian white noise).

The cluster power output in Figure 6 shows that the active power model predictive controller has better robustness than the conventional PD strategy. The additional random noise does influence the predicted available power and the proportional coefficient for each unit. Because of the open loop characteristic of the conventional strategy, the cluster power output cannot be guaranteed with no difference when tracking the set-point, and the deviations from the predicted available power will influence the active power output directly. As for the proposed model predictive controller, the predicted available power mainly provides a constrained feasible interval for each unit power set-point instead of adding the influence directly. The predicted error will be mitigated by the spare capacity of each unit in the cluster without causing any significant deviations of the cluster power output. The constrained feasible interval and power output for each unit in Figure 7 shows such a feature very clearly. In Figure 7, the blue curves are the generator active power output for four units, and the black curves are the constrained feasible intervals by the time-varying (19). Once the cluster power set-point changed, the feasible interval reaches its maximum range and gradually reduces if the cluster power set-point keeps still until meeting its minimum range after 30s. During the time-varying region, power output of units 3 and 4 are gradually increasing; meanwhile, units 1 and 2 cut out several power outputs to maintain the adequate spare regulation capacity after the fast tracking of

power set-points. The comparison about the unit power output between the conventional strategy and the proposed strategy is shown in Figure 8.

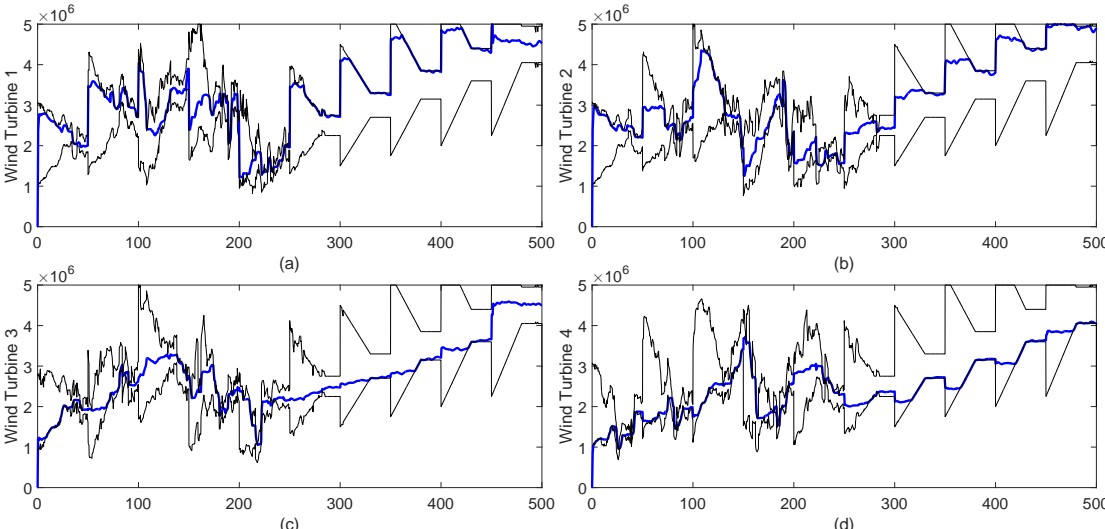

**Figure 7.** Four-units cluster robustness test—the active power output under the proposed controller for each wind turbine, sub-figures a, b, c, d are the active power output and the constraints under the proposed strategy for wind turbine 1, 2, 3, 4 respectively.

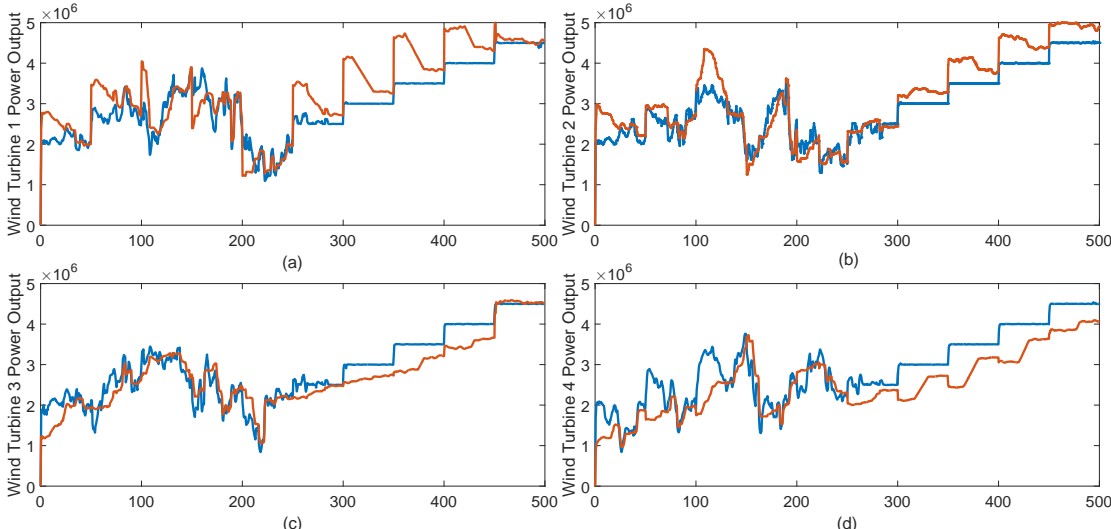

**Figure 8.** Four-units cluster robustness test—comparison between the proposed strategy and the conventional strategy, sub-figures a, b, c, d are the active power output comparison between the proposed strategy and conventional strategy for wind turbine 1, 2, 3, 4 respectively.

Moreover, the selected cluster is also tested under the irregular time-varying power set-point, where the cluster power output and each units power output are plotted in Figures 9 and 10. Obviously, the cluster power output under the proposed controller follows the set-point change quickly and almost coincides with the power set-point. In Figure 10, the slow time-scale units served as a mainstay under the frequent set-point change, and the fast time-scale units are giving full play of their own regulation flexility to support the cluster active power demand.

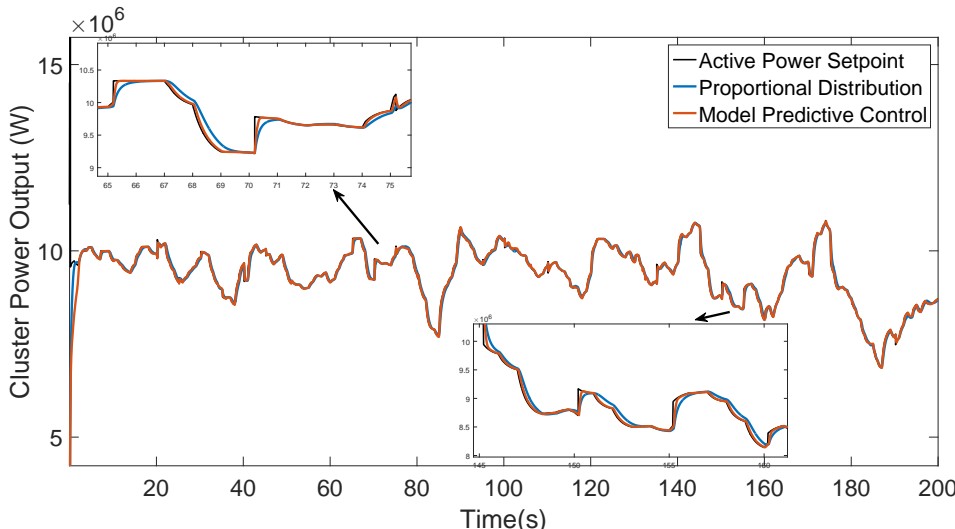

**Figure 9.** Four-units cluster robustness test—cluster power output under varying power demand.

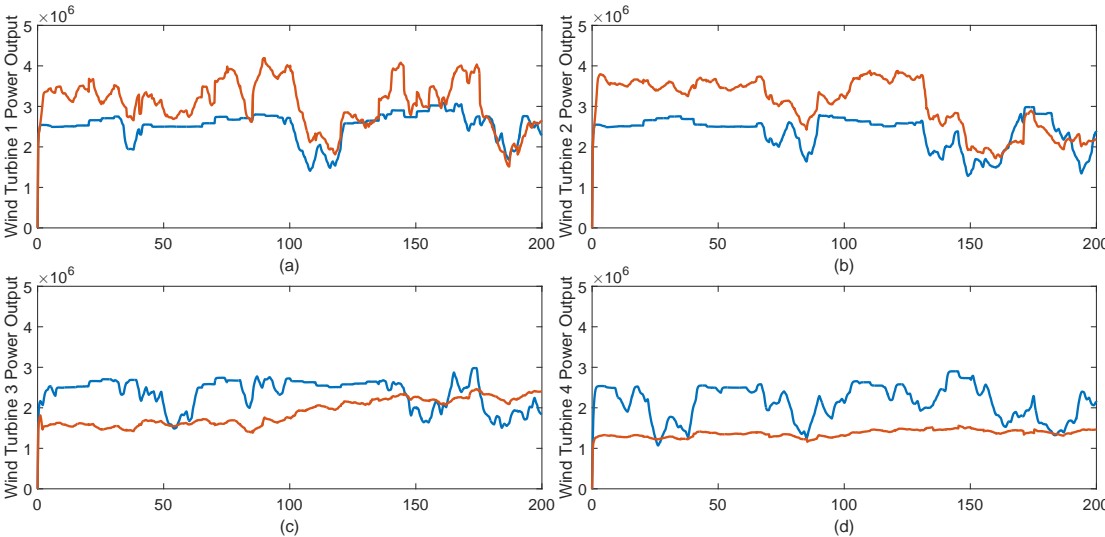

**Figure 10.** Four-units cluster robustness test—comparison between the proposed strategy and the conventional strategy under varying power demand, sub-figures a, b, c, d are the detailed comparison for wind turbine 1, 2, 3, 4 respectively.

### 6.2. Four-Units Cluster Active Frequency Support Test

Similar to the above cluster, four units in one wind farm are selected as a flexible scheduling cluster to support the wind farm power output and frequency support ability. The four units are included in a 100 MW wind farm connected in a IEEE standard 14-bus electric power system. The wind farm contains 20 wind turbines and each unit has a capacity of 5 MW. The entire wind farm is divided into five clusters according to the connection of electric power transmission lines. Each power transmission line is connected to four units which can be scheduled as a cluster. The topological structure and unit connection mode of the wind farm are set as in Figure 11, where each unit is connected by a 1.5 km electric power transmission line. The four units used in above simulations are connected to one power transmission line as a cluster. Another failure cluster is founded with four units of which the actual wind turbines are replaced by amplifiers with static gain of 1 to simulate the active power transient change of the wind farm. The wind farm controller superimposes two signals with equal amplitude but opposite direction on the basic active power set-point and sends them to the selected cluster and failure cluster, respectively, and the selected cluster acts as required to test the active power

and frequency support ability for the wind farm. The other 12 conventional units are working by
following a constant power set-point to avoid the unnecessary interference.

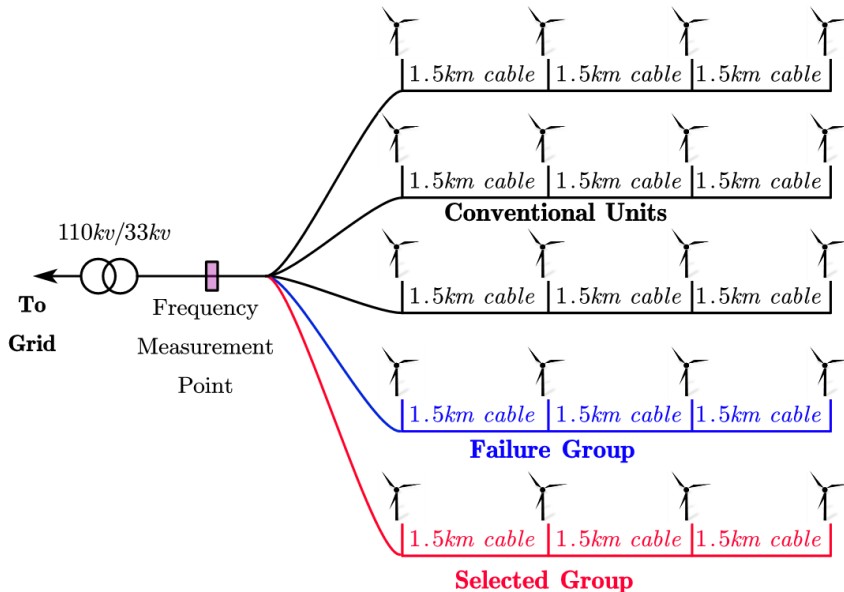

**Figure 11.** Units connection structure in the tested wind farm.

The selected cluster is simulated under two types of test singles. The first active power set-point
is the step change single to test the cluster active power and frequency support ability under a unit
emergency cut out/start situation. Under the first condition, several units are selected to cut out or
join up to provide a step change active power signal and the frequency response of the wind farm are
plotted as in Figure 12. The cluster power set-point is the same as the above simulations.

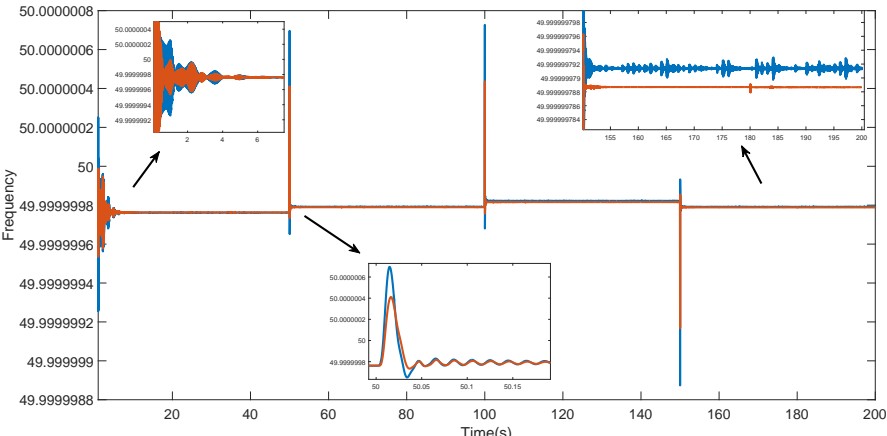

**Figure 12.** Wind farm exit node frequency response.

In Figure 12, the blue curves are the wind farm frequency response under the PD control strategy,
while the orange one is the corresponding to the proposed strategy. Clearly, under the proposed
strategy, the wind farm frequency response is much more stable than the conventional strategy.
When the wind farm active power changes, the frequency is stabilized quickly at the initial stage.
Once the power set-point step changed, the frequency under the proposed controller has a lower
overshoot and faster adjusting time than the conventional controller. In addition, during the time
range when the cluster power set-point remains still, the frequency response under the proposed

controller has little fluctuation, which indicates that the cluster inner disturbance caused by natural wind turbulence and units' inertial difference can be significantly restrained by the proposed controller.

Note that, in Figure 12, the wind farm frequency responses under different cluster control methods show a similar trend but not completely the same. This is due to the low frequency oscillation of power grid, which is still a challenge problem in the power system. The proposed strategy mainly focuses on unit active power control and optimization within a cluster concept, which cannot handle the low frequency oscillation problem.

### 6.3. Four-Units under a Wind Farm

In the above simulations, the tested cluster shows a great stabilized power tracking ability under the proposed active power control strategy. However, the capacity of four tested units will be a limit that the wind farm usually contains much more units than the tested cluster, where the cluster mainly provides a middle layer between the wind farm and each unit. Thus, the wind farm is controlled as Figure 13 to test the regulation ability of the selected cluster ability. In the wind farm, the selected four units are selected as the tested cluster and the other 16 units form another conventional cluster. The wind farm power demand is split into two parts based on the maximum capacity of two clusters. The conventional cluster is controlled by following the PD strategy, and the tested cluster is controlled under the proposed strategy or the PD strategy to verify the influence of the tested cluster on the wind farm power output. The wind farm is tested under both step-change power demand and time-varying demand, and the farm power output are shown in Figures 14 and 15.

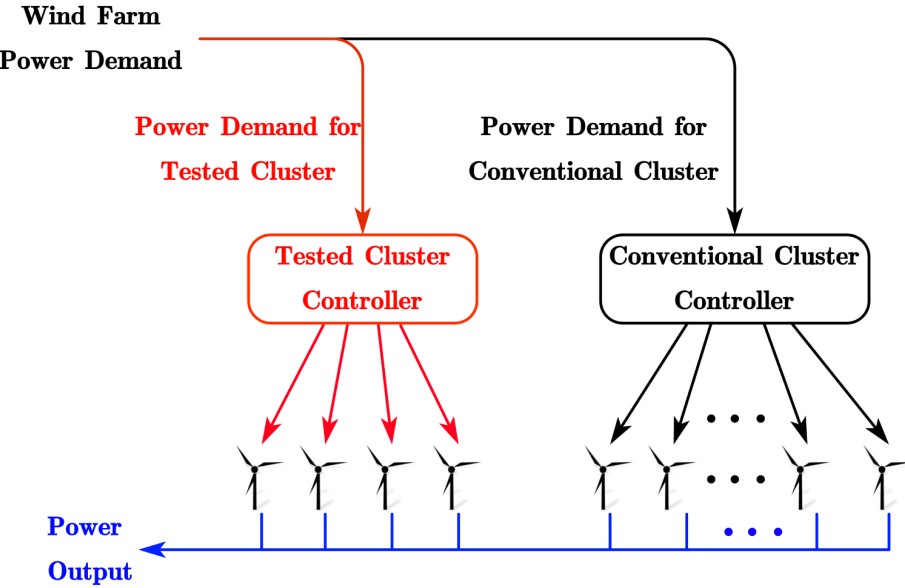

**Figure 13.** Wind farm control structure.

Clearly, when the tested cluster is controlled under the proposed MPC strategy, the wind farm active power is improved in both tracking ability and output stability. The power output in Figure 15 tracks the power demand more quickly than the PD strategy. The wind farm power output under the step demand in Figure 14 also shows the effectiveness when the selected cluster is controlled by the proposed strategy and the conventional cluster follows the PD strategy. Although the power fluctuation from the conventional cluster still brings oscillations to the farm power output, the improvement is still significant.

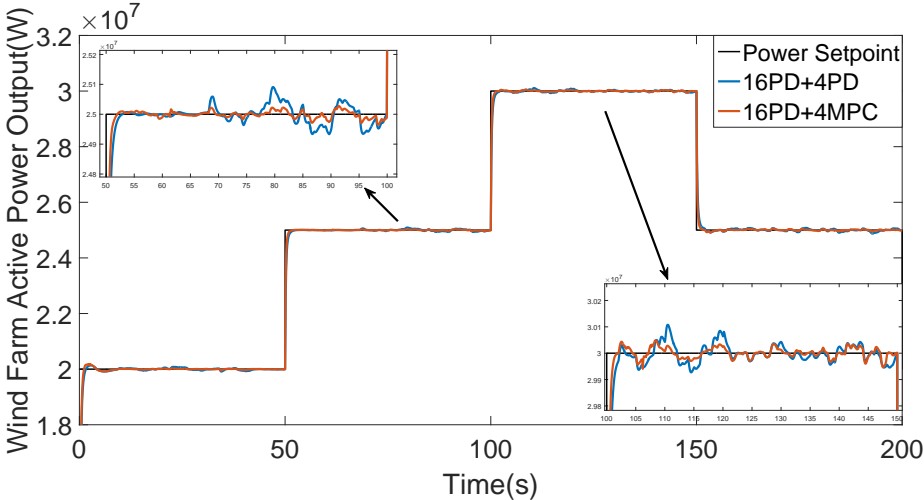

**Figure 14.** Wind farm power output under a step changes power demand.

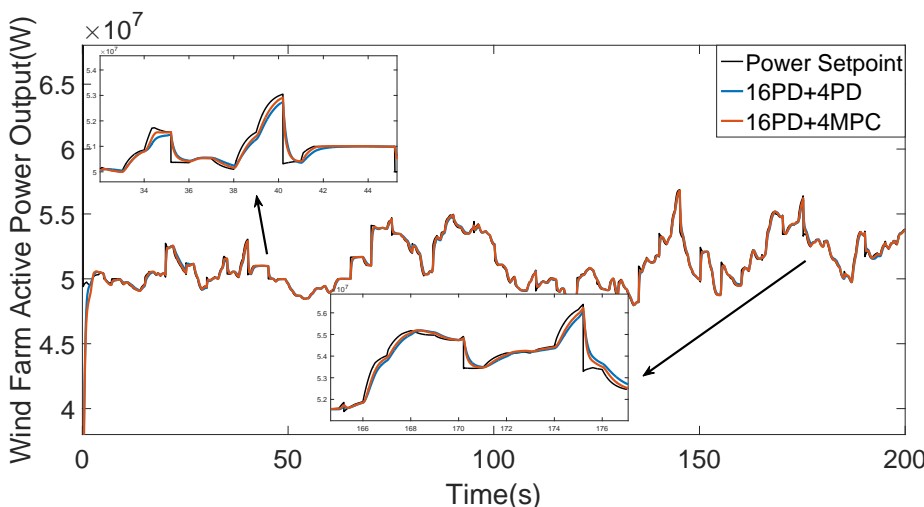

**Figure 15.** Wind farm power output under a time varying power demand.

To show the improvement in a intuitive way, the integrated absolute error (IAE) is selected as the quantitative index to show the above improvement when using the proposed strategy, which is calculated as follows:

$$IAE = \int_0^T |e(t)| \, dt,$$

in which $e(t)$ is the tracking error between the power set-point and power output for the wind farm or the wind turbine cluster; $T$ is the simulation time. The corresponding result is shown in Table 1.

**Table 1.** IAE(integrated absolute error).

| Object | IAE_Step | IAE_TimeVarying |
|---|---|---|
| Cluster_4PD | 6.75335 | 19.54523 |
| Farm_4PD | 13.4768788 | 51.32480 |
| Cluster_4MPC | 1.721556 | 10.02838 |
| Farm_4MPC | 9.477333 | 43.02592 |

In Table 1, the IAE_step is the IAE index for the wind farm and wind turbine cluster under a step change set-point as shown in Figure 14, and IAE_TimeVarying is the result under the time varying power demand in Figure 15. Clearly, the improvement for a cluster level is significant, but when

considering the wind farm level, the capacity of the tested cluster is still a limit to further improve the wind farm power output.

On the other hand, the proposed strategy mainly considers the active power distribution and frequency support under a cluster level. Essentially, it belongs to the secondary frequency modulation under the wind farm scheduling, which can be easily extended to primary frequency modulation only needs a suitable frequency-power droop curve for the cluster. However, it is not appropriate to choose a large-scale wind farm as a cluster and apply the proposed strategy directly. Under our simulations, the increase of the units number in the cluster will lead to the prolongation of the optimization problem solving time, and the problem caused by modeling deviation will also be serious. All 20 units of the wind farm are selected and tested under a step change set-point and the electric power output and frequency responses are shown as in Figures 16 and 17.

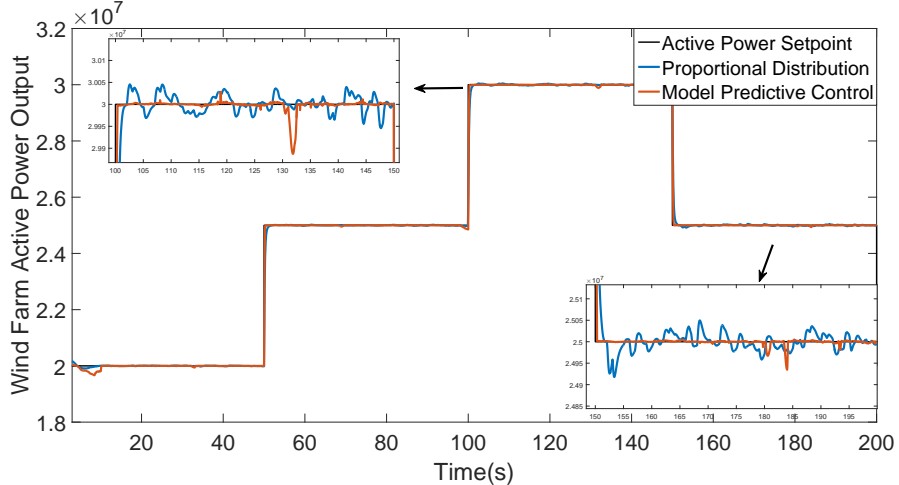

**Figure 16.** Wind farm active power output.

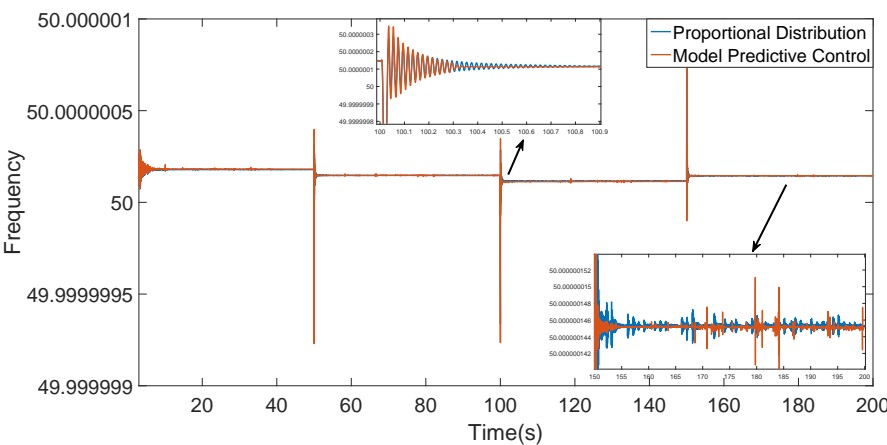

**Figure 17.** Wind farm exit node frequency response (20 units under model predictive control).

In Figure 16, the wind farm active power output under the proposed controller still keeps a fast tracking and stable characteristic, but there still exist several unnecessary pulses due to modeling deviations. Several concussion attenuations under the conventional strategy are changed into pulses under the proposed controller. Although the output frequency oscillation can be quickly suppressed, such frequency pulses still have adverse impact on the electric grid. In a word, the proposed controller works efficiently under a small cluster with several units, but the increase in the number of units will amplify the internal disturbances and cause unnecessary concussions. Another thing to note is that, in this paper, the wake effects between wind turbines are ignored. During the available power predictive process, each wind turbine predicts the available power for itself to make sure the

power set-point for each unit can be achieved, and further wind-turbine cluster power distribution is implemented based on the detailed available constraints for each units. Under such circumstances, once the later units are affected by wake from front units, the predicted available power for the infected wind turbine will be maintained at the correct power, the difference caused by wake can be obtained during the active power control process and the proposed strategy still can achieve proper control effect. Furthermore, once the wind turbines in a wind farm can be clustered into groups, one group can be regarded as an equivalent wind turbine through appropriate parameter aggregation methods, and the proposed strategy can be used as the active power distribution method from the wind farm to equivalent wind turbines. Further research could focus on the application of the proposed strategy to a wind farm with other necessary improvements like wake effect and so on to achieve the wind farm active power optimization.

## 7. Conclusions

This paper discussed the flexibility problem of the active power distribution from a wind-turbine cluster level. The wind turbine power tracking dynamic characteristic is simplified as an inertial link, which can reflect the difference between wind turbine power tracking inertial time constant. A model predictive controller is proposed to control and optimize the active power set-point from the wind-turbine cluster to the inner wind turbines. In particular, the time-varying boundary constraints and relaxed constraint for total power set-point are proposed, that is, to maintain the slow time-scale units keeping a basic active power output and the fast time-scale units playing important role when cluster power set-point changes. Several simulations and comparisons prove the effectiveness and advancement of the proposed strategy. However, the detailed simulations indicate that there still exists improvement space for applying the proposed strategy to the entire wind farm. Future research could be concentrated on reducing the solving time for a high-order model predictive controller and eliminating the internal disturbance within the cluster for better active power output tracking and frequency support.

**Author Contributions:** Z.C. and C.Q. performed the experiments, Z.C. and Z.L. wrote the paper, J.L. and Z.L. revise and editing the paper.

**Funding:** This work was funded by the National Natural Science Foundation of China (No. U1766204), the Fundamental Research Funds for the Central Universities (No. 2018ZD05, 2018BJ0116), the Research on the intelligent control technology of the wind turbine of China Guodian United Power Technology Co., Ltd. (No. GPOD17001), the Development and application of key technologies for grid-friendly intelligent wind farm energy management system of Guodian New Energy Technology Research Institute Co., Ltd. (No. GJNY-18-22).

**Conflicts of Interest:** The authors declare no conflict of interest.

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
