# Peer review of "Variable-Constrained Model Predictive Control of Coordinated Active Power Distribution for Wind-Turbine Cluster"

_applsci, doi:10.3390/app9010112_

Reviewer 1 Report

The whole paper suffers because it is never made clear what actual real-life problem it is really addressing, what the important issues are, and what performance criteria are important.

Page 2: ADMM is not explained. Terminology generally is poorly defined.

This paper appears to completely ignore wake effects. There is one reference to SimWindFarm, so maybe there are wake effects, but the methodology needs to be explained in much more detail so that the reader understands what has been modelled and to what level of detail and comprehensiveness.

How is the available power calculated in equation 8? There is completely inadequate explanation of the modelling used.

"the inertia time constants of simulated wind turbines are identified" - again, there is no explanation of the turbine model used for this.

Why do the turbines all have such very different time constants? Do they vary with operating conditions?

Page 7 refers to the "turbine start/stop problem" but this is not important. Turbines can generate zero power for a while without having to shut down.

What is the controller time step? The ringing in Figure 16 indicates that it is very fast. Surely there is no need to control active power so fast.

Author Response

We would like to express our sincere appreciation to the reviewer for their careful assessments and constructive comments on our submission, particularly the time being spent. We take the reviewer's view very seriously and have made every possible effort in order to address the concerns raised by the editor and reviewer. The detailed modification can be referred to the revision. Answers to the reviewer as given in the subsidiary document.

Reviewer 2 Report

The article is interesting, relevant and reasonably well written. It does however base its results on rather simplified wind turbine and wind turbine controller models, which are reduced to first order LTI systems with remarkably short time constants. This leads to the turbines achieving MW-level step changes in a split second. This is, in fact, not realistic for multi-MW wind turbines, which require considerably smoother generator torque changes to remain mechanically viable. I would suggest the Authors use the publicly available FAST model of the NREL 5MW reference turbine to somehow calibrate their inertial link models.

Author Response

(The authors gave the same response as above.)

Reviewer 3 Report

This paper proposes the active power tracking method by deciding power command using model predictive framework. I have some doubts.

Firstly, the model describing the wind turbine dynamics (in eq. (2)) is too simple to account the wind turbine dynamics. This assumption could be very serious problem since the wind turbine has many nonlinear dynamics and practically could not be defined as first order model. Thus, authors should convince that this assumption can be reasonable.

Secondly, why the time varying constraints are needed as in eq. (10)? Authors described the reason that to make the small time-scale units taking the leading part in the fast tracking. But, is it not enough to use the cost function in eq. (6)?  Too much constraints could make it infeasible solution problem when we consider other cases such as multiple severe frequency events.

Author Response

We would like to express our sincere appreciation to the reviewer for their careful assessments and constructive comments on our submission, particularly the time being spent. We take the reviewer's view very seriously and have made every possible effort in order to address the concerns raised by the editor and reviewer. The detailed modification can be referred to the revision. Answers to the reviewer as given in the subsidiary document.

Round  2

Reviewer 1 Report

Thank you for you modifications. The explanations are more complete now. I still think that you should mention wake effects somewhere, even if just to say that you are ignoring them and to explain why the results still have some value despite this.

Author Response

Thanks to the reviewer for this valuable suggestion, appropriate explanations are added in this revision to clear that the proposed strategy ignores wake effects.The detailed modification is explained in the appendix file and can be also referred in the manuscript.

Reviewer 2 Report

None

Author Response

We are grateful to the reviewer for the positive opinion. In this revision, we

have tried our best to improve the clarity of our results.

Reviewer 3 Report

I am satisfied about the authors reply and recommending this paper to be published in current form.

Author Response

(The authors gave the same response as above.)
